# Development of a Predictive Dosing Nomogram to Achieve PK/PD Targets of Amikacin Initial Dose in Critically Ill Patients: A Non-Parametric Approach

**DOI:** 10.3390/antibiotics12010123

**Published:** 2023-01-09

**Authors:** Anne Coste, Ronan Bellouard, Guillaume Deslandes, Laurence Jalin, Claire Roger, Séverine Ansart, Eric Dailly, Cédric Bretonnière, Matthieu Grégoire

**Affiliations:** 1Service de Maladies Infectieuses et Tropicales, CHU de Brest, 29200 Brest, France; 2Cibles et Médicaments des Infections et de l’Immunité, 9 IICiMed, UR1155, Nantes Université, 44000 Nantes, France; 3Laboratoire de Traitement de l’Information Médicale, INSERM, UMR1101, Brest Université, 29200 Brest, France; 4Service de Pharmacologie Clinique, CHU Nantes, 44000 Nantes, France; 5Unité de Neuro-Anesthésie-Réanimation, Groupe Hospitalier Pitié-Salpétrière, AP-HP, 75013 Paris, France; 6Département d’anesthésie et réanimation, douleur et médecine d’urgence, CHU Carémeau, 30029 Nîmes, France; 7UR UM 103 IMAGINE, Faculté de Médecine, Montpellier Université, 30029 Nîmes, France; 8Service des Soins Intensifs de Pneumologie, CHU Nantes, 44000 Nantes, France

**Keywords:** amikacin, ICU, pharmacokinetics, dosing, nomograms

## Abstract

French guidelines recommend reaching an amikacin concentration of ≥8 × MIC 1 h after beginning infusion (C_1h_), with MIC = 8 mg/L for probabilistic therapy. We aimed to elaborate a nomogram guiding clinicians in choosing the right first amikacin dose for ICU patients in septic shock. A total of 138 patients with 407 observations were prospectively recruited. A population pharmacokinetic model was built using a non-parametric, non-linear mixed-effects approach. The total body weight (TBW) influenced the central compartment volume, and the glomerular filtration rate (according to the CKD–EPI formula) influenced its clearance. A dosing nomogram was produced using Monte Carlo simulations of the amikacin amount needed to achieve a C_1h_ ≥ 8 × MIC. The dosing nomogram recommended amikacin doses from 1700 mg to 4200 mg and from 28 mg/kg to 49 mg/kg depending on the patient’s TBW and renal clearance. However, a C_through_ ≤ 2.5 mg/L 24 h and 48 h after an optimal dose of amikacin was obtained with probabilities of 0.20 and 0.81, respectively. Doses ≥ 30 mg/kg are required to achieve a C_1h_ ≥ 8 × MIC with MIC = 8 mg/L. Targeting a MIC = 8 mg/L should depend on local ecology.

## 1. Introduction

Amikacin is commonly used in association with a beta-lactam agent in patients with septic shock to broaden the spectrum of the initial antimicrobial therapy and to potentiate bactericidal effects [1,2]. Its use has been associated with better outcomes in patients with a high risk of death [3,4]. A rapid discontinuation of the combination therapy is recommended after bacterial documentation [5], and a single infusion of amikacin is often given to a patient [6]. Clinical success is associated with a ratio of the plasma concentration one hour after the beginning of a short-infusion (C_1h_) to a minimal inhibitory concentration (MIC) higher than eight to ten [7,8,9,10]. Nephrotoxicity is associated with a residual concentration of amikacin ≥2.5 mg/L. French guidelines recommend a 30 mg/kg of total body weight (TBW) dose with a 30-min infusion for intensive care unit (ICU) patients in order to achieve an amikacin C_1h_ eight to ten times above the MIC of the targeted bacteria [11]. For obese patients, the adapted body weight (ABW) is preferred over TBW. According to the EUCAST breakpoint for *Enterobacterales*, a MIC of 8 mg/L should be targeted for probabilistic treatments [12], resulting in C_1h_ targets from 64 to 80 mg/L. This easy method to calculate the optimal dose of amikacin is widely used, but the PK–PD C_1h_ target is only achieved in 60 to 82% cases [6,13,14]. This may be explained by modified pharmacokinetic properties and the broad inter- and intraindividual variability of aminoglycoside exposure in ICU patients, which justify individualized dosing regimens and, when possible, model-informed precision dosing approaches [15,16]. There is a need to develop a pragmatic and individualized dosing strategy to maximize the probability of target attainment right from the first dose of amikacin in ICU patients.

Many pharmacokinetic models have been elaborated to describe amikacin concentrations [17,18,19]. However, most of them are combined with Bayesian forecasting and cannot be easily applied for initial dosing. Dosing nomograms have been used for a long time [20] and are still developed for various antibiotics [21,22,23,24] due to their proven efficacy [25] and simplicity of use. The aim of this study was to design a pharmacokinetic model based on easily available patient characteristics and plasma concentrations to predict the amikacin peak. We then developed a nomogram to determine the optimal first dose of amikacin to be administered based on a patient’s characteristics.

## 2. Results

### 2.1. Patients’ Characteristics and Amikacin Concentrations

A total of 138 patients (91 patients from PICAMI and 47 patients from AMINO2) corresponding to 407 amikacin measurements were included in the analysis. All patients presented septic shock. The patients’ characteristics are summarized in Table 1. The median [quartile 1 (Q1), quartile 3 (Q3)] first dose of amikacin was 2000 [1660–2400] mg, corresponding to 27.7 [22.6–29.9] mg/kg of TBW. This dose was associated with a median [Q1, Q3] first C_1h_ of 67.5 [54.6–86.7] mg/L. Only 78 (56.5%) and 44 (31.9%) patients achieved target C_1h_ values ≥ 64 mg/L and 80 mg/L, respectively.

### 2.2. Population PK Modeling and Evaluation with Pmetrics

A two-compartment model best described the observed data with first-order elimination. More complex structural models did not significantly improve the Akaike information criterion (AIC), population bias or imprecision. The intercompartmental clearance (Q) was fixed to its mean (5.43 h^−1^), as this parameter could not be accurately estimated due to a high variance. Among the studied covariates, two were found to have an influence on the model parameters (*p* < 0.05): TBW had an influence on the central compartment volume (V) with a linear relationship, and the glomerular filtration rate (GFR) estimated with the Chronic Kidney Disease–Epidemiology Collaboration (CKD–EPI) formula (eGFR) had an influence on the central compartment clearance (CL), with a power relationship. A gamma error model with a starting value of 3 was chosen, and the values for error coefficients C0, C1, C2, and C3 were 1.25, 0.05, 0, and 0, respectively, for the SD polynomial. The final value of gamma was 1.5, which indicated good-quality data. The final structural model is represented in Figure 1. PK parameter estimates are summarized in Table 2.

Diagnostic plots are presented in Figure 2. The bias and imprecision were 0.16 and 5.79, respectively, for population predictions and −0.04 and 0.80, respectively, for individual predictions. Residual plots for population predictions showed an even distribution of weighted residual errors over the concentration range and over time. The VPC revealed that the prediction was consistent with the observations (Figure 3).

### 2.3. Monte Carlo Simulations and Dosing Nomograms

The range of the eGFR for the nomogram was fixed from 20 to 135 mL/min/1.73 m^2^ (10th to 90th percentile of the observed eGFR in the studied population). A dosing nomogram for amikacin is presented in Figure 4 and shows the dose per kg of TBW of amikacin needed to achieve a C_1h_ ≥ 64 mg/L with a probability of target attainment (PTA) of 0.9 for a median TBW of 76.5 kg and for the 10th and 90th percentiles of the observed population TBW (52 and 125 kg, respectively). The optimal dose varied from 1700 mg to 4200 mg and 28.4 mg/kg to 49.0 mg/kg of TBW depending on the TBW and eGFR.

A C_through_ ≤ 2.5 mg/L 24 h and 48 h after the beginning of the amikacin infusion of an optimal calculated dose was obtained in simulated patients of median TBW and eGFR with a PTA of 0.20 and 0.81, respectively.

## 3. Discussion

In ICU patients, immediately choosing an adapted first dose of amikacin is crucial. The objective of this work was to provide a simple tool to optimize this choice in daily practice. We conceived a dosing nomogram to choose the ideal dose of amikacin needed to achieve the defined PK–PD target, depending on two patients’ easily obtained characteristics: TBW and eGFR.

A non-parametric pharmacokinetic modeling approach was used [26]. The model showed good performance and adequately described the observed concentrations. Most blood samples were collected at the same time, 1 h after the beginning of injection, but this issue was compensated for by the high number of observations and the use of a population modeling approach. The eGFR and TBW were identified as covariates influencing amikacin CL and V, respectively.

GFR estimates are included in most amikacin pharmacokinetic models [16,17]. However, this crucial parameter is not considered for choosing the first dose of amikacin in official guidelines, probably because its impact on amikacin C_1h_ is deemed to be minor. However, the C_1h_ of amikacin is not the maximum concentration, as it is measured 30 min after the end of a 30-min amikacin infusion. At the time of measurement, clearance has already occurred for one hour. We know that ICU patients can experience acute kidney injury (AKI) and augmented renal clearance (defined as a creatinine clearance ≥ 130 mL/min) [27]. In our cohort study, these cases accounted for a quarter of patients (15.2% of eGFR < 30 mL/min and 11.6% of augmented renal clearance). We showed that the first dose of amikacin needed to achieve a C_1h_ ≥ 64 mg/L varied from 30 mg/kg to 40 mg/kg in patients of an average weight, depending on the eGFR. This could explain some of the reported failures to reach PK–PD targets [13,28,29].

In this study, the GFR estimated with the CKD–EPI formula best increased model precision. The CKD–EPI formula was developed in 2009 [30] and is nowadays commonly used to estimate the glomerular filtration rate. The precision of the CKD–EPI equation is consistent for underweight and obese patients [31]. However, its accuracy in ICU patients has been challenged, especially in patients with AKI [32]. The calculation of exact creatinine clearance requires urine collection, which was not performed in this study. This limitation reflects real-life conditions, as urine analysis is often unavailable during the initial management of patient with septic shock. Consequently, we produced a dosing nomogram for patients with a GFR estimated with the CKD–EPI formula as ≥20 mL/min.

French guidelines recommend adjusting the amikacin dose depending on TBW with a linear relationship, except in obese patients where ABW is used [10]. Pathophysiologic changes in obesity affect most hydrophilic medications, and many authors recommend estimating the volume of distribution using ABW in obese patients [33]. In our study, 30% of patients were obese (body mass index ≥ 30 kg/m^2^). However, TBW was chosen over ABW because of its better predictive performance. Elaborating nomograms based on TBW also presents the advantage to being easy to use for clinicians. The predicted optimal dose per kg of TBW for obese patients was found to be lower than that for patients of average weight. The difference was more significant for underweight patients who required a much higher dose-to-weight ratio. Indeed, the complexity of the relationship between weight and optimal amikacin dose cannot be reduced to a simple linear relationship without a loss of efficiency.

We built a nomogram showing the lowest first dose of amikacin to administer to ICU patients with septic shock to reach a 64 mg/L C_1h_ target with a PTA above 0.9. Our nomogram showed that the recommended relative dose of 30 mg/kg by French guidelines is not enough for most patients. For example, a patient with an eGFR of 100 mL/min/1.73 m^2^ and a TBW of 50 kg would need a 40 mg/kg dose. Moreover, as dose calculation is based on TBW according to our model rather than ABW according to French guidelines, overweight patients would receive higher doses. For a patient with a median eGFR of 70 mL/min/1.73 m^2^ and a TBW of 125 kg, the 30 mg/kg recommended dose amounts to 3750 mg and 2800 mg when dose calculation is based on TBW and ABW, respectively.

These high doses may raise safety concerns, as amikacin can cause nephrotoxicity and ototoxicity. A study linked amikacin toxicity with C_1h_ [13]_,_ but this finding has not been confirmed by other studies [29]. In the current state of knowledge, no upper limit for C_1h_ has been defined. Rather, nephrotoxicity is associated with amikacin C_through_ [34]. Its incidence was shown to dramatically decrease with once-daily regimens [35]. French guidelines recommend targeting a C_through_ < 2.5 mg/L. According to our simulations, this target cannot be achieved with our recommended dose 24 h after the last amikacin administration in most cases. French guidelines recommend decreasing the infusion frequency instead of reducing the amikacin dose to avoid amikacin accumulation in renal tissues. Indeed, both nephrotoxicity and ototoxicity have been linked with a total amikacin dose > 9 g [36], and animal studies have associated nephrotoxicity with amikacin accumulation in tissues [37]. In the ICU, the total amikacin dose is of little concern because clinicians usually give a single dose of amikacin before switching to another drug. However, more studies, such as animal safety studies, are needed to determine if nephrotoxicity can occur after a single high dose of amikacin.

As emphasized in a recent study, the targeted C_1h_ depends on the MIC of the targeted pathogen [38]. French guidelines recommend considering a MIC of 8 mg/L for probabilistic therapy targeting *Enterobacterales*; consequently, the C_1h_ target is 64 mg/L. In order to reach probabilistic C_1h_ targets, clinicians could use either a high dose of amikacin, as calculated in our study, or target a lower MIC. Some studies have shown that despite failing to achieve a priori PK–PD targets, a 30 mg/kg dose enabled the attainment a posteriori PK–PD targets due to the low MIC of the encountered pathogens [6,28,39]. Aiming pathogens with a MIC of 4 mg/L rather than 8 mg/L for probabilistic therapy is associated with a more reachable C_1h_ and consequently C_through_ targets. Most wild-type pathogens have a MIC ≤ 4 mg/L [40]. However, amikacin-acquired resistance is seen worldwide, especially in the ICU [41]. The EUCAST defined new clinical breakpoints for amikacin due to the recent use of once-daily high doses and the acknowledgment of their use in combination with other active therapy. *Enterobacterales* and *Pseudomonas aeruginosa* are now defined as susceptible to amikacin if they have MIC values ≤ 8 and 16 mg/L, respectively [12]. Our results question the choice of breakpoint MICs and the definition of amikacin susceptibility. The decision to target a lower MIC for probabilistic therapy should be made according to local epidemiology.

This work suffered from several limitations. First, our nomogram has not yet been prospectively validated on an external cohort, which limits its safe use for now. A bicentric prospective cohort study is planned. Second, our predictions were based on our population characteristics. Despite a multicentric design, this study included patients from a single country and our results may not be generalized to other populations, especially for patients with extremely high body weight or renal clearance.

In conclusion, we produced a dosing nomogram to help clinicians choose the optimal first dose of amikacin in ICU patients. The optimal dose depends on patient eGFR and TBW. High doses ≥ 30 mg/kg are often required if high MICs are targeted (more or equal to 8 mg/L). It is therefore necessary to assess one’s local ecology before aiming at this target. Finally, the use of these high doses needs to be assessed in terms of safety, given that a single dose is usually administered.

## 4. Materials and Methods

### 4.1. Patient Population and Data Collection

Data were obtained from two prospective French cohorts: (i) the AMINO2 study [28], including patients from the Nîmes University Hospital between October 2014 and February 2015, and (ii) the PICAMI study [6], including patients from the Nantes University Hospital between July 2014 and November 2016. Both studies were approved by local review boards. We included patients over 18 years old and treated for sepsis in the ICU with combination therapy including aminoglycosides at the day of enrolment.

According to French guidelines, patients received a 30-min intravenous infusion of amikacin diluted in 50 mL of NaCl 0.9%. Plasma concentration sampling was performed 30 (±15) minutes after the end of infusion (C_1h_).

For all patients at the day of inclusion, the following kinds of data were collected: age, sex, height, and SAPS II [42]. On every day of amikacin infusion, the following data were additionally collected: weight, serum creatinine, total proteinemia, albuminemia, renal replacement therapy, vasopressors use, and invasive mechanical ventilation. Body mass index and body surface area (BSA) were calculated with the Dubois formula [43], ideal body weight (IBW) was calculated with the Lorentz formula [44], ABW was calculated with the Traynor formula [45], estimated GFR was indexed by BSA according to the CKD–EPI formula [30] (eGFR, mL/min/1.73 m^2^), and estimated GFR was calculated with the Cockroft and Gault and the Modification of Diet in Renal Disease (MDRD) formulas.

### 4.2. Bacterial Susceptibility Testing and Amikacin Quantification

Bacterial antibiotic susceptibilities, including amikacin MICs, were determined using a VITEK 2 automated system (bioMérieux, Marcy-l’Etoile, France). Amikacin concentrations were measured using fluorescence polarization automated immunoassays (Cobas 8.000 kit, Roche, Basel, Switzerland). The limits of quantifications were 0.8 mg/L in the AMINO2 study and 2.5 mg/L in the PICAMI study.

### 4.3. Population Pharmacokinetics Analysis

The population PK model was built using the non-parametric adaptive grid algorithm with the *Pmetrics* package for *R* (version 4.0.4, Laboratory of Applied Pharmacokinetics, Los Angeles, CA, USA) [46,47].

#### 4.3.1. Base Model

One-compartment, two-compartment and three-compartment structural models were initially tested without covariates to determine the best-fitting structural model. The selection of the most appropriate model was based on the AIC (an estimate of the likelihood penalized by the number of parameters in the model), population bias and imprecision calculations (the mean weighted error of predictions minus observations and the bias-adjusted mean weighted squared error of predictions minus observations, respectively), and diagnostic plots of observed concentrations versus population-predicted and individually predicted concentrations and of weighted residual error versus time or individual predictions.

An additive gamma:error=SD×γ
and multiplicative lambda:$$error={\left(SD^{2}+\lambda^{2}\right)}^{0.5}$$
error models, where SD is the standard deviation of each observation and γ and λ represent process noise (such as model misspecification and sampling time uncertainty), were tested. SD was modelled by a polynomial equation:$$C_{0}+C_{1}\times \left[obs\right]+C_{2}\times {\left[obs\right]}^{2}+C_{3}\times {\left[obs\right]}^{3}$$
where [obs] is the observed concentration.

#### 4.3.2. Covariate Model

The relationship between the model parameters and the different covariates was evaluated using stepwise linear regression, AIC, and the visual assessment of plots of the model parameters against covariates. The covariates were then selected using forward and backward stepwise selection on the Chi^2^ test of the objective function. Continuous covariates were integrated into the structural model using either a linear
$$P=P1\times P2\times \left(\frac{COV}{COV_{median}}\right),$$
exponential
$$P=P1\times e^{P2\times \left(\frac{COV}{COV_{median}}\right)},$$
power
$$P=P1\times {\left(\frac{COV}{COV_{median}}\right)}^{P2},$$
or allometric for weight only
$$P=P1\times {\left(\frac{COV}{COV_{median}}\right)}^{P2}$$
relationship, where P, P1, and P2 are parameters; COV is the covariate value; and $COV_{median}$ is the covariate median in the dataset. Binary covariates were integrated into the structural model using either a linear
P=P1+P2×COV,
or exponential
$$P=P1\times P2^{COV}$$
relationship. Covariates that improved the model according to the AIC, bias, and imprecision were integrated into the final model. Parameter ranges were initially set wide and then narrowed to increase the density of support points in the pertinent range. The process was iterated until no further improvement to the model was observed.

#### 4.3.3. Model Evaluation

The evaluation of the final model was conducted using graphical methods. Visual predictive checks (VPCs) were performed using the *vpc* package for R. For each patient in the dataset, 1000 Monte Carlo simulations were performed. Medians, 5th percentile, and 95th percentile of the observed and simulated concentrations were visually compared. Concordance was visually checked, and bias and imprecision were calculated.

### 4.4. PTA and Dosing Nomogram

Monte Carlo simulations (*n* = 1000), based on the parameters of the structural model, were generated from patient profiles with a large panel of covariate values (eGFR and TBW) between the 10th and 90th percentiles of the population. For each of these profiles, exposure to amikacin was assessed for doses ranging from 1300 mg to 5800 mg administered in a 30-min intravenous infusion.

Targeted amikacin concentrations were defined as $C_{1h}>8\times MIC$ when MIC = 8 mg/L (64 mg/L) according to French guidelines [11] and EUCAST breakpoints [12]. Optimal doses to achieve this PK–PD target were determined for various eGFR values and for TBW values equal to the 10th, median, and 90th percentiles of the observed weight in the dataset. Exploratory simulations were conducted to determine the PTA of C_through_ target ≤ 2.5 mg/L 24 h and 48 h from the beginning of the infusion of optimal doses.

To conceive the dosing nomogram, the lowest dose required to achieve a PTA of 0.9 1 h after beginning the infusion was reported (GraphPad Prism version 8.0.2, San Diego, CA, USA).

## Figures and Tables

**Figure 1 antibiotics-12-00123-f001:**
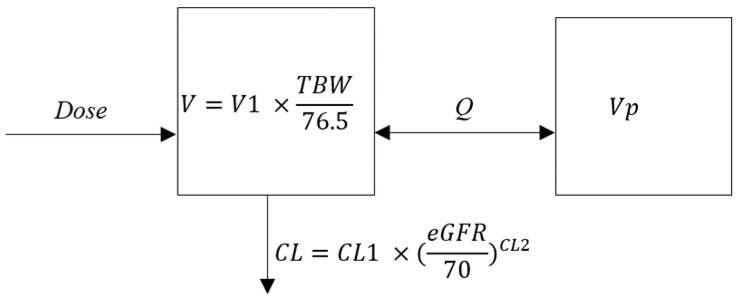
Structural model. V: central compartment volume; Q: intercompartmental clearance; Vp: peripheral compartment volume; CL: central compartment clearance; V1, CL1, and CL2: parameters; TBW: total body weight; eGFR: estimated glomerular filtration rate.

**Figure 2 antibiotics-12-00123-f002:**
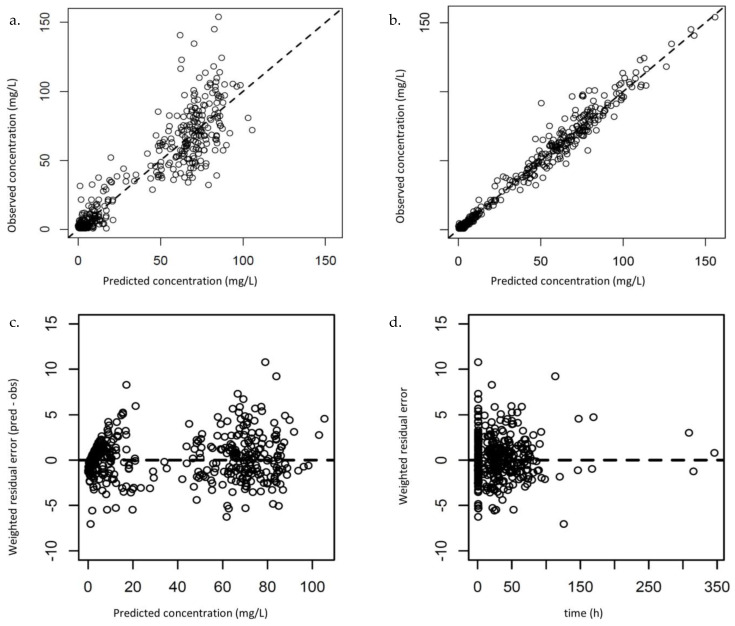
Diagnostic plots. Observed versus population-predicted amikacin concentrations (**a**) and individual-predicted amikacin concentrations (**b**). Weighted residual error plotted against predicted concentrations (**c**) and time (**d**) for population predictions. pred: predictions; obs: observations.

**Figure 3 antibiotics-12-00123-f003:**
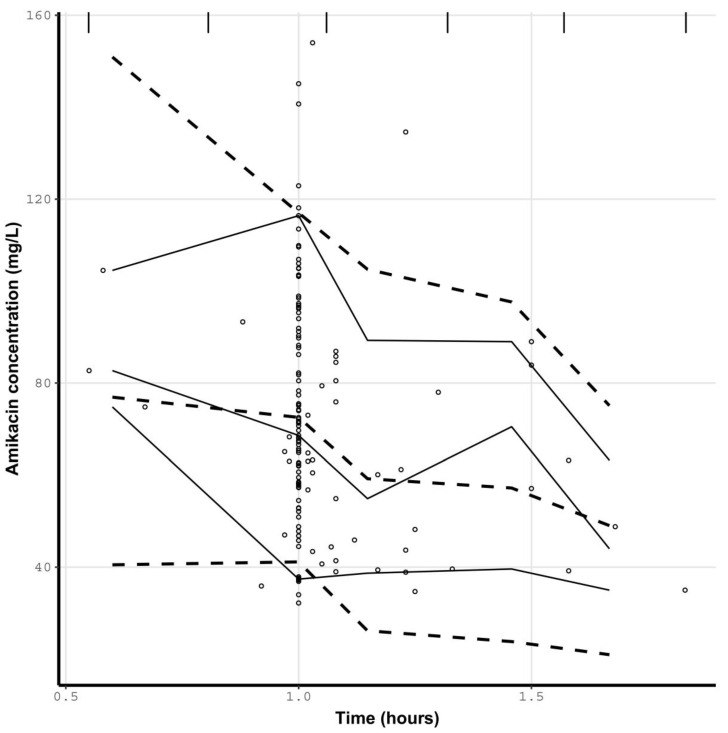
Visual predictive checks of amikacin concentrations against time. Open circles are observed amikacin concentrations. Solid lines represent the 5th, 50th, and 95th percentiles for observed concentrations. Dashed lines represent the 5th, 50th, and 95th percentiles for simulated concentrations. The vertical lines at the top of the plot are bin separators.

**Figure 4 antibiotics-12-00123-f004:**
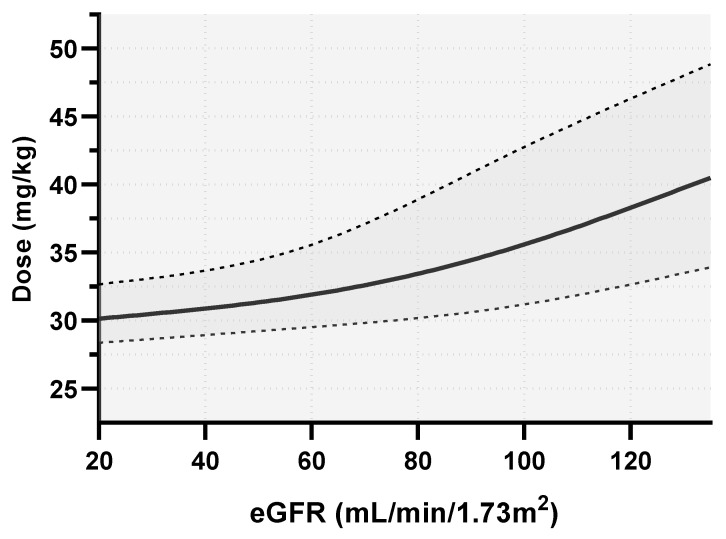
Nomogram of the median first dose of amikacin to be administered to attain a C_1h_ ≥ 64 mg/L in 90% of the studied population for a TBW of 76.5 kg (median value of the studied population, full line), a TBW of 52 kg (10th percentile, top dashed line), and a TBW of 125 kg (90th percentile, bottom dashed line).

**Table 1 antibiotics-12-00123-t001:** Patient’s characteristics.

Patient’s Characteristics	Median [Q1–Q3] or *n* (%)
Age (years)	62 [49–72]
Sex (male)	90 (65.2)
Total body weight (kg)	76.5 [62.0–88.5]
Height (cm)	170 [163–177]
Body mass index (kg/m^2^)	25.6 [21.9–31.4]
Body mass index ≥ 30 kg/m^2^, *n* (%)	41 (29.7)
Body surface area (m^2^)	1.89 [1.72–2.03]
SAPS II	41 [31–56]
Total proteinemia (g/L)	55 [48–62]
Albuminemia (g/L)	26.0 [20.6–30.8]
Serum creatinine (µmol/L)	91 [61–171]
eGFR (CKD–EPI formula, mL/min/1.73 m^2^)	70.0 [41.9–111.5]
Renal replacement therapy	9 (6.5)
Vasopressors use	79 (57.3)
Invasive mechanical ventilation	94 (68.1)

eGFR: estimated glomerular filtration rate; CKD–EPI: Chronic Kidney Disease–Epidemiology Collaboration; SAPS II: Simplified Acute Physiology Score II.

**Table 2 antibiotics-12-00123-t002:** Population parameter estimates.

Parameter ^1^	Median (95% CI)	MAWD (95% CI)	Range
CL1	4.41 (3.84–5.27)	1.24 (0.64–1.99)	0.01–15.00
CL2	0.72 (0.57–0.92)	0.29 (0.15–0.56)	0.01–10.00
V1	20.40 (16.58–26.49)	5.45 (2.58–8.88)	0.01–60.00
Vp	16.32 (13.71–23.99)	7.20 (3.94–14.95)	0.10–100.00

CI: confidence interval of the estimates; MAWD: median absolute weighted deviation, used as an estimate of the variance for a nonparametric distribution; range: interval of values set before the run. ^1^ In the model, CL = CL1 × (eGFR/70)^CL2^, where CL is the elimination rate constant from the central compartment (per hour) and eGFR/70 is the estimated glomerular filtration rate (milliliters per minute) normalized to the population median. V = V1 × (TBW/76.5), where V is the volume of the central compartment (liters) and TBW/76.5 is the total body weight (kilograms) normalized to the population median. Q was fixed to 5.43 h^−1^.

## Data Availability

Not applicable.

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
