# Peer review of "Development of a Predictive Dosing Nomogram to Achieve PK/PD Targets of Amikacin Initial Dose in Critically Ill Patients: A Non-Parametric Approach"

_antibiotics, 2023, doi:10.3390/antibiotics12010123_

Round 1

Reviewer 1 Report

The purpose of this study was to develop a nomogram to determine the optimal first dose of amikacin to be administered based on patient’s characteristics. The content of this research is more systematic and appropriate, but the innovation is slightly insufficient.

Major comments:

1 The blood samples were mostly collected at 1 h post dose, which might bring some bias to the estimation of the PK parameters.

2 Is there any upper limits for the target C1h, such as the toxicity level? When a percentage of 90% of the study population were exposed to C1h ≥ 64 mg/L, it probably had a lot of concentrations much higher than 64 mg/L.

3 Figure 4 should be improved to make the readers easily to find the first dose. Maybe some more auxiliary dashed lines in the figure.

Reviewer 2 Report

Development of a predictive dosing nomogram to achieve 2 PK/PD targets of amikacin initial dose in critically ill patients: 3 a non-parametric approach – novel idea but several flaws in the methodology require substantial amendments

Achieving a target of , C1h target is 64 mg/L is too ambitious to be achieved clinically and if the MIC of 8 mg/L (although guidelines list as susceptible), clinicians should be very hesitant to use this drug and should seek alternatives. This is because the target is not feasible to achieve without exposing the patient to known and unknown toxicities which were not studied for – we need animal safety studies before thinking to use in humans

Aminoglycosides dosing should be based on adjusted body weight not total body weight specifically in obese (almost 30 % of the population had BMI > 30)

Sample median age not listed

Several levels were taken from same patients (investigators did not mention how this confounder was eliminated)

Median eGFR was 70.0 [41.9-111.5] and given patients in septic shock should have had AKI, where the once daily dosing is contraindicated (although he 25%  were 41.9, using eGFR in patients with AKI is not adequate as their estimated eGFR usually < 10 mL/min/m2 -further 6.5% utilized RRT) which could have led to flaws in the results

Figure 4, the produced nomogram suggest that the first dose of 25 mg/kg is needed if the patient had eGFR of 20 to achieve a level of 64 mg/L???? Is that reasonable? How long this dose will be staying in the body (septic shock with AKI) – this could last for a week if no RRT utilized with such high level leading to further toxicities (unknown)

The authors stated that “French guidelines 186 recommend targeting a Cthrough < 2.5 mg/L”, however this study does not predict when we can reach this level (for redosing purpose) especially big chunk of the patients were in ICU with Septic shock (i.e. AKI)

Round 2

Reviewer 1 Report

The manuscript is improved, but it still needs some minor revision. The followings are my comments.

1. At line 296-297, the power for the allomentric model was fixed to 1.5. Could the author explain the reason to fixed 1.5? It is usually fixed to 0.75 for the clearance or 1 for the apparent volume. Accutally, the author finally estimated the power factor, which is near 0.75 in Table 2. 

2. In Figure 1, in the equation CL=CL1*(eGFR/70)^CL2, a pair of parentheses should be used to the eGFR/70.

3. Table 2, in the note under the table, "...WT/76.5 is the total body weight..." should be "...TBW/76.5 is the total body weight...". It should be the same as that in the equation.

Author Response

Dear reviewer,

Thank you very much for your last comments. These were typing errors, that are now corrected.

Sincerely yours

Reviewer 2 Report

Thank you for the responses, however such high doses can not be suggested without trials on animal models to check safety. I understand that the authors cited older studies, however, they can not recommend such high doses based on PK models (the known adverse events of nephro and oto are the ones known, however, how can we guarantee there will not be different side effects

Author Response

Dear reviewer,

We made consequent changes to the conclusion paragraph to make sure people understand the need to assess the safety of single high doses of amikacin before using such high doses.

Sincerely yours,